# Nuts, Energy Balance and Body Weight

**DOI:** 10.3390/nu15051162

**Published:** 2023-02-25

**Authors:** David J. Baer, Michelle Dalton, John Blundell, Graham Finlayson, Frank B. Hu

**Affiliations:** 1US Department of Agriculture, Agricultural Research Service, Beltsville, MD 20705, USA; 2School of Psychology, Leeds Trinity University, Leeds LS18 5HD, UK; 3School of Psychology, Appetite Control and Energy Balance Research Group, University of Leeds, Leeds LS2 9JT, UK; 4Department of Nutrition, Harvard T.H. Chan School of Public Health; Boston, MA 02115, USA

**Keywords:** energy, calories, mastication, appetite, food intake, body weight, obesity, nuts

## Abstract

Over several decades, the health benefits of consuming nuts have been investigated, resulting in a large body of evidence that nuts can reduce the risk of chronic diseases. The consumption of nuts, being a higher-fat plant food, is restricted by some in order to minimize weight gain. In this review, we discuss several factors related to energy intake from nuts, including food matrix and its impact on digestibility, and the role of nuts in regulating appetite. We review the data from randomized controlled trials and observational studies conducted to examine the relationship between nut intake and body weight or body mass index. Consistently, the evidence from RCTs and observational cohorts indicates that higher nut consumption does not cause greater weight gain; rather, nuts may be beneficial for weight control and prevention of long-term weight gain. Multiple mechanisms likely contribute to these findings, including aspects of nut composition which affect nutrient and energy availability as well as satiety signaling.

## 1. Introduction

Achieving and maintaining a healthy body weight is a difficult goal for many individuals. Obesity is a global health issue. According to the WHO [1], in 2016, almost 40% of the world’s population were overweight and over 10% had obesity (among adults aged 18 years and older). Obesity is largely preventable, and at the simplest level, it is a matter of appropriate energy balance. To lose weight, energy intake must be less than energy expenditure, and to maintain weight, energy intake and expenditure must be equal. Within the constructs of this simple energy balance problem, there are many interdependent and complex factors that make body weight maintenance difficult. These factors include factors related to food and macronutrient composition, as well as the food matrix and energy availability. Appetite regulation is even more complex with a multitude of organ systems involved in making decisions multiple times, every day, about what to eat, when to eat, when to stop eating and how much to eat. Ultimately, interactions amongst food and its consumption determines energy intake.

In the early 1990s, research was beginning to show beneficial health effects associated with nut consumption [2]. With the emerging evidence, the 1995 Dietary Guidelines for Americans mentions including nuts in the diet but cautioned that foods, including nuts, high in fat should be used sparingly [3]. In the subsequent decades, much additional research has been conducted to better understand the health benefits of nuts, including the role nuts play in body weight maintenance. Herein, we review the state of the science in regard to how the food matrix of nuts effects energy availability, how nuts effect ingestive behavior and the literature on the relationship between nut intake and body weight maintenance.

## 2. Energy Availability

The plant cell wall significantly effects the bioavailability of energy and nutrients from nuts. Plant cell walls are complex extracellular matrices containing cellulose, hemicellulose, pectin and some proteins (usually enzymes that play a role in cell wall integrity). Lignin, which are polymers made of phenylpropanoid units, are additionally found in secondary cell walls [4]. Together, the cell wall components provide structural integrity for the plant, encapsulate the cell membrane to protect the individual cell and play a role in water and nutrient transport. Nutritionally, the plant cell wall is the source of dietary fiber which is resistant to mammalian digestive enzymes. Microbial anerobic fermentation or physical breakage of the plant cell wall is needed in order to release the cellular contents and make those contents available for absorption.

Among the first studies to report on the bioaccessibility of fat from nuts was a study of peanuts, peanut butter and peanut oil [5]. In this small study, subjects consumed each treatment, feces were collected and daily fecal fat excretion was determined. Consumption of peanuts resulted in increased fecal fat excretion compared to peanut butter and oil, and consumption of peanut butter resulted in increased excretion of fecal fat compared to peanut oil [5]. As seen in other studies, the absorption of lipids from oils is generally quite high in humans [6,7]. However, when the same fat is consumed in the plant matrix, bioaccessibility is decreased, resulting with increased fecal fat excretion.

### 2.1. Effect of Processing and Mastication on Almond Lipid Bioaccessibility

Microscopic analyses of nuts demonstrate that several factors contribute to breakage of the plant cell wall. Prior to that breakage, these microscopic analyses show that the lipid vacuoles remain intact, encased by the plant cell wall. Mastication is one factor which influences lipid bioaccessibility. In one study, subjects masticated and expectorated natural (unroasted) or roasted almonds [8]. Post-mastication, there were differences in the particle size distribution between the natural and roasted almonds, with larger particles (1700 to >3350 µm) being more prevalent in the natural almond samples and smaller particles (<1700 µm) being more prevalent in the roasted almond samples [8]. Lipid was identified on the surface of ruptured cells. In smaller particles (approximately 250 µm), free lipid was identified in all areas of the particles, not just the surface. In this study, roasting effected lipid bioaccessibility, which was greater in the roasted almonds compared to the natural almonds, and the higher lipid bioaccessibility was related to the increased proportion of smaller particles observed in the roasted almonds [8].

In a study of different forms of almonds (natural, roasted, roasted diced and almond butter (made from roasted almonds)), following simulated oral digestion, particle size distribution was similar for natural, roasted and chopped almonds (most particles having a size ≥1000 µm), whereas the particle size distribution of the almond butter resulted in mostly smaller particles (<850 µm) [9].

#### 2.1.1. Effect of Roasting

Roasting changes the physical properties of almonds, and these changes contribute to the degree of cell ruptures. Using three-point bending to determine fracture force (N) at load failure, roasted almonds required less force for load failure than natural almonds [10]. The hardiness of roasted almonds, quantified by maximum force (N) required for failure during uniaxial compression, was also lower than natural almonds [10]. Upon fracture, 8-bit, binary digitized photos were used to quantify particle area and total number of fragments. The median particle area was smaller for roasted almonds compared to natural almonds, whereas there was a greater number of particles from roasted compared to natural almonds. The physical changes associated with roasting impact lipid bioaccessibility by increasing the ratio of surface area to volume and making more cellular contents available for digestion and absorption.

#### 2.1.2. Effect of Mastication

Mastication is one of the physical processes which plays an important role in bioaccessibility of fat and energy. In a study of controlled mastication, subjects were provided 5 g of natural almonds and instructed to chew them for 10, 25 or 40 times and then expectorated [11]. The number of particles recovered was measured post-mastication, with more particles recovered after 10 chews compared to 25 or 40 chews [11]. Moreover, the mean particle size of the recovered particles was larger after 10 chews than 25 or 40 chews [11]. In a separate study with these subjects, they were allowed to chew the almonds and swallow after 10, 25 or 40 chews. Fecal samples were collected and fecal energy and fat were measured. Fecal energy and fat extraction were higher after 10 chews compared to 25 or 40 chews [11]. In this study, chewing almonds 10 times resulted in differences from chewing almonds for 25 or 40 times, but additional chewing of almonds beyond 25 times did not significantly change the particle size distribution, fecal energy or fat excretion.

#### 2.1.3. Observations with Walnuts and Pistachios

Much of the research on the effects of processing and mastication has been conducted with almonds. One study [12] used walnuts (unsalted pieces) and pistachios (roasted) in addition to almonds (roasted and salted), and focused on in vitro gastrointestinal digestion. In undigested samples, walnuts had thinner cell walls compared to almonds and pistachios, whereas pistachios had smaller oil bodies than walnuts and almonds. Transmission electron microscopy revealed that in walnuts and almonds, the lipid was stored in a single, dense agglomerate, whereas the lipid in pistachios was observed in smaller and dispersed droplets within the cell [12]. Following mastication and in vitro digestion, cell walls from all nuts showed fissures and free lipids in the extracellular space. Thus, the effect of mastication and digestion (in vitro) of walnuts and pistachios also results in the breakage of cell walls, the release of lipids and increased bioaccessibility.

### 2.2. History of Determining the Energy Value of Foods

For food labeling purposes, the metabolizable energy value of the food is typically used. Metabolizable energy is the gross energy of the food corrected for energy losses in feces and urine. Gross energy of food, feces and urine are measured by bomb calorimetry. For food labeling, direct measures of metabolizable energy are not performed, but rather, the metabolizable energy is estimated using energy density factors which represent the energy, adjusted for incomplete digestion. The energy density factors, commonly known as the Atwater general factors, were based on research conducted by Atwater and colleagues [13]. Based on these studies, Atwater proposed that the metabolizable energy value of protein, fat and carbohydrate could be estimated as 4, 9 and 4 kcal/g, respectively. These factors were further refined based on food groups, more targeted to improve digestibility estimates of macronutrients. These refined energy density factors are commonly known as the Atwater specific factors.

While there is no evidence that Atwater performed studies with nuts, Jaffa performed studies with walnuts, Brazil nuts, pecans and almonds [14]. These studies were conducted with two to three men, involved simple diets usually containing a few items and lasted for a few days. While the intention of Jaffa’s and Atwater’s research was to provide information on the energy and nutrient availability of mixed diets, their results have been applied to individual foods. After the seminal work of Atwater and Jaffa, few additional studies have been reported focusing on measuring the energy value of individual foods or simple diets. While the state-of-the-art nature of Atwater’s work has been the foundation for nutrition labeling, the approach is not without limitations. Some of the limitations have been reviewed [15] and include the small sample size, short duration of collections and measurement errors.

### 2.3. Recent Measures of the Metabolizable Energy Value of Nuts

In order to better measure the metabolizable energy value of an individual food while it is being consumed as part of a mixed diet, Novotny developed an approach [16] that overcomes the limitations of the Atwater approach [15]. Briefly, this approach requires a pair of diets—one without the food of interest and the other an identical diet with the food of interest. Using this approach, the metabolizable energy value of pistachios [17], almonds [16], walnuts [18] and cashews [19] was investigated. Additionally, a study of different forms of almonds was conducted [10].

In all of these studies, the measured metabolizable energy value of the nuts was lower than the energy value calculated using Atwater general or specific factors. The difference between the measured and calculated metabolizable energy values were 6% for pistachios (whole, lighted roasted and lightly salted) [17], 19% for almonds (whole, unroasted, unsalted) [16], 21% for walnuts (pieces) [18] and 6% for cashews (whole, roasted, lightly salted) [19]. In all of these studies, the amount of nuts included in the diet was 42 g/day (1.5 oz/day), and this amount was selected to be consistent with the US FDA qualified health claim for nuts [20,21]. Furthermore, with the study of pistachios and almonds, a second amount of 84 g/day was used to investigate dose effects. There was no change in the measured metabolizable energy value between the two doses [16,17].

## 3. Appetite as a Complex System

The regulation of appetite is complex and influenced by various biological, nutritional, physical and social factors. Humans are omnivores, allowing them to make food choices from a wide range of available options, but this versatility can also be a challenge. Appetite can be broadly divided into tonic and episodic components, which are generally represented by the drive to eat and food choice behavior. The key determinants of the drive to eat are the body’s lean mass and resting metabolic rate, but these are unrelated to food choice [22]. Food hedonics, or the experienced pleasure derived from eating food, has a major influence on food choice. The consumption of chosen foods inhibit the drive to eat through the processes of satiation and satiety, which form part of the Satiety Cascade [23]. Satiety is the post-prandial inhibitory component of appetite control and is mediated by complex physiological processes. There is huge individual variability in the way people experience satiety, and the strength of satiety is heavily influenced by the diet selected.

The effect of nuts on appetite can be assessed by scientifically investigating their effects on the processes of satiation and satiety in relation to an individual’s pattern of satiety control. When people freely consume nuts as part of their diet, either within or between meals, it is important to enquire what effects this will have on their overall energy intake and their pattern of food consumption. To investigate this issue, it is necessary to understand some features of the appetite system and the mechanisms that mediate the effects of ingested foods.

### 3.1. Appetite and Satiety (and Satiation)

Appetite is not the opposite of satiety; rather, satiety is an important component of appetite control. Appetite encompasses various processes that influence food consumption, including the initiation of eating and the duration and termination of eating episodes. Such processes include the level of hunger, food availability and choice, food hedonics, psychological traits, situational factors and social factors. Satiety refers to the reduction in hunger and eating following a meal. Satiation, on the other hand, is the termination of a meal and affects meal size. The period after a meal involves a complex series of physiological events in the digestive tract, including gastric activity and hormone release, which control the digestion and absorption of nutrients. These physiological events depend on the type of the foods consumed in the diet. As they accompany the state of satiety, they are often referred to as satiety signals. Whether these satiety signals are biomarkers of satiety or the cause of it is an area of debate in the field. However, it can be concluded that there is no single unique satiety signal [24]. As satiety is an inhibitory process, it plays an important role in determining how much food is eaten and one’s levels of hunger. Therefore, satiety may potentially influence body weight by either permitting or preventing overconsumption [25,26]. Weak satiety is seen as a major factor in obesity, while intensifying satiety through certain foods or drugs may support weight loss [27,28,29].

### 3.2. Foods and the Satiety Cascade

As omnivores, human beings have the capability to consume a vast variety of foods from around the world, leading to a range of unique dietary patterns. The foods chosen impact the levels of satiation and satiety felt. This phenomenon can be explained through the idea of the ‘Satiety Cascade’ (Figure 1) [23]. The satiety cascade provides a framework to understand the mechanisms involved in the short-term control of eating behavior.

The satiety cascade distinguishes between satiation and satiety and illustrates how a variety of signals, such as those arising from sensory, cognitive, post-ingestive and post-absorptive processes, affect the frequency and size of meals. The processes of satiation control meal size through their effect on the duration and termination of an eating episode. These processes, along with the nutritional content of the food consumed, determine the amount of energy consumed during the eating episode. Once the meal is finished, the desire to eat is temporarily suppressed by the physiological effects of the consumed food, especially in the stomach, and the hormones released by the gastrointestinal system during the digestion and absorption of food.

### 3.3. The Nature of Satiety Signals

After eating, the sensation of fullness (satiety) is produced by several features of the foods consumed, including volume, weight, sensory features (taste, texture, mouthfeel), enjoyment, appearance, nutritional composition, non-nutritional elements (such as fiber) and packaging/labeling. Therefore, satiety is a result of the combined effects of various components of the food consumed. Many studies have aimed to determine the specific characteristics of foods that have the most significant effect on satiety. Understanding these factors is crucial for the food industry in creating foods that can regulate hunger and enhance the feeling of fullness. There is evidence to suggest that high protein and fiber levels can increase satiety. However, energy density is a crucial factor, with low-energy-density diets producing stronger feelings of satiety [31].

Since the first investigations of satiety, it has been believed that the impact of food composition is influenced by post-meal physiological responses. These responses involve alterations in gastric distension, digestion and emptying, as well as the release of gastrointestinal peptides including cholecystokinin (CCK), glucagon-like peptide (GLP-1), peptide YY (PYY), insulin and others. For a long time, CCK was considered to be the sole satiety signal. It is important to note that all these peptides play important roles in the body’s management of food through digestion and absorption processes, such as slowing down gastric emptying and releasing bile for fat emulsification. As a result, their impact on satiety may be secondary to their other functions. It remains a topic of discussion whether gut peptides are markers or the actual cause of satiety. The fact that different foods may have similar effects on satiety but produce distinct physiological profiles suggests that there is no uniform pattern behind satiety and that the same level of satiety may be linked to different post-prandial physiological changes [24]. In recent years, there has been considerable interest in the post-prandial physiological effects of raw foods compared with highly and ultra-processed foods [32].

### 3.4. A Note on (the Role of) Food Hedonics

Food is a reliable source of pleasure for most people, and the reward derived from food plays an important role in the initiation, maintenance and termination of an eating episode, in part through interaction with processes involved in hunger and satiety. Food hedonics is more than simply liking the taste of food or the experience of pleasure. Non-human animal research has demonstrated that the brain structures underpinning food hedonics comprise dissociable affective and motivational subcomponents, termed ‘liking’ and ‘wanting’, respectively. Liking refers to the sensory pleasure experienced while eating a food and is generated by the binding of opioids to specialized clusters of neurons in the reward pathway, particularly in the Nacc shell. Wanting refers to the process that assigns motivational value to finding and consuming a food and is mediated by the release of dopamine (DA) from the ventral tegmental area (VTA) to the nucleus accumbens (Nacc) and amygdala [33,34]. In human appetite research, the terms “liking” and “wanting” for food are often seen as explicit subjective states that correspond to their everyday meanings. Liking refers to the enjoyment of the sensory qualities of food that give it its hedonic impact, while wanting refers to a subjective state of desire or craving. People are generally good at estimating and reporting their liking for food, but are often inaccurate in their assessment of their implicit wanting for food, meaning why they are attracted to or craving a particular food over another [35,36].

After food is consumed, the sensory aspects of the food are registered by both cognitive and sensory processes before it is swallowed. Highly palatable food stimulates the reward pathways in the brain, causing the release of dopamine and endorphins. These reward pathways have connections to the hypothalamus, which triggers the release of hunger-inducing peptides such as NPY and orexins and suppresses the release of satiety-inducing peptides such as insulin, leptin and cholecystokinin. Thus, the consumption of highly palatable food can result in overeating, as the drive to eat is motivated by pleasure rather than actual hunger. The interplay between the hedonic and homeostatic systems of appetite regulation contributes to the overall pattern of eating behavior, and in an environment that promotes obesity, the hedonic drive to eat may have a stronger impact on food consumption compared to homeostatic mechanisms [29,37].

### 3.5. Individual Variability in Appetite Control and the Low Satiety Phenotype

The range of factors that contribute to a person’s susceptibility to overconsume (and eventually weight gain and obesity) can include their genetics, physical and psychological characteristics, lifestyle habits and surrounding food and activity environment. Decades of research have pinpointed many aspects of the typical Western lifestyle and diet that interact with these factors, making it easier for people to overeat and gain weight. However, not everyone in a ‘westernized’ environment overconsumes food, and it is unlikely that one single factor can account for why some are more vulnerable than others; this has implications for appetite control and the prevention of weight gain.

One approach to characterize individual susceptibility is through the identification and characterization of phenotypes. One such phenotype may be characterized by a weakened satiety response to food, which has been proposed as a possible marker of susceptibility to overeating [38,39,40]. Research has shown that under controlled conditions, appetite sensations are a valid and reliable method for measuring the subjective motivation to eat [41]. However, not everyone reports a good relationship between their sensations of appetite (hunger and fullness) and their eating behavior. A weakened satiety response to the feeling of satiety may play a role in a lack of control over one’s appetite. Individuals with such a reduced response to food are referred to as the “low satiety phenotype” [42]. The low satiety phenotype has largely been observed in people with obesity, but evidence suggests that a weakened satiety response to food may lead individuals to be vulnerable to future weight gain. Research examining the low satiety phenotype has demonstrated that it is characterized by increased Three Factor Eating Questionnaire disinhibition and hunger scores, lower levels of craving control, greater food wanting and increased energy intake under laboratory conditions [42,43,44]. In terms of weight management, studies have shown that individuals with a low satiety responsiveness tend to lose less weight, experience smaller decreases in abdominal fat, report lower control over cravings and face more challenges in sticking to a diet compared to those with a high satiety responsiveness [44,45,46].

### 3.6. Nuts and Appetite Control: A Case Study with Almonds

Snacking between meals is a common behavior [47], and snack foods make a significant contribution to total daily food intake [48]. Snack foods are often characterized as being low in nutritional quality, primarily comprising fats and carbohydrates [49] that contribute to overconsumption. However, research suggests that frequent snacking can promote feelings of satiety throughout the day, which results in overall lower daily energy intake [50]. This suggests that snacking behavior itself is not undesirable and may present an opportunity for the addition of healthy foods, such as nuts, into the diet [51]. A recent meta-analysis of randomized clinical trials found that regular consumption of nuts was associated with increased daily energy intake and lower hunger but had no effect on weight or feelings of fullness [52]. The increase in daily energy intake was lower than the amount of energy consumed from the nuts, which may be due to the lower amount of available energy from nuts following digestion [16,53].

Almonds are a natural food product that are high in protein and fiber as well as fat, but lower in metabolizable energy compared to the predicted levels (using Atwater factors) [16]. It is well established that proteins and fibers have prominent effects on appetite control [54,55], and since they act via different mechanisms, their effects may be additive. The unique structural properties and macronutrient composition of almonds may be beneficial for the control of hunger, strength of satiety and subsequent energy intake relative to other foods. The addition of almonds to a meal has been shown to increase satiety and decrease blood glucose concentrations in those with and without impaired glucose tolerance [56,57,58]. Furthermore, when consumed as a snack, almonds have been shown to reduce feelings of hunger and desire to eat [59,60]. The consumption of almonds as a snack does not seem to cause an increase in total daily energy intake [61] or result in significant weight change over time [62,63,64]. A recent study compared the effect of consuming almonds as a mid-morning snack compared to an energy- and weight-matched comparator snack (crackers) and a zero-energy, weight-matched control (water) on measures of subjective appetite, food intake and food hedonics. It was found that overall hunger was lower in the almonds condition, and almonds were more satiating than the crackers (Figure 2). There was also a reduction in implicit wanting for high-fat food following almond consumption suggesting a beneficial effect on hedonic hunger. Further to this, participants’ perceptions of the almonds were favorable, with almonds being perceived as healthy, filling and good for weight management [61].

### 3.7. Nuts and the Low Satiety Phenotype

As outlined above, the low satiety phenotype is characterized by higher levels of hunger across the day, greater overall energy intake, increased liking and wanting for food, and poorer weight loss outcomes following structured weight management programs [42,43,44,45,46]. Foods that promote satiety have the potential to support individuals (in general and perhaps in particular those with a weakened satiety response to food) to control their appetite, eat fewer calories and manage their weight [65]. Research suggests that even when matched for calories, not all foods provide the same level of satiety [66], and a hierarchy of macronutrient satiating power has been established, with foods that are high in protein and fiber and low in energy density being more satiating [54,67,68,69]. The unique structural properties and macronutrient composition of nuts may be beneficial for the control of hunger, strength of satiety and subsequent energy intake relative to other foods; therefore, the consumption of nuts may support those with low satiety responsiveness in improving their appetite control.

Hollingworth [70] compared the effect of consuming almonds as a mid-morning snack compared to an energy- and weight-matched comparator snack food (crackers) on satiating efficiency, energy intake and feelings of hunger and fullness across the day in the low satiety phenotype compared to the high satiety phenotype. They found that almonds had a greater satiating efficiency, measured using the satiety quotient, in the low satiety phenotype compared to the comparator snack. In addition, when compared to the comparator food, almonds were perceived as being healthier, more filling and more favorable for weight management. Expectations about the satiating potential of food has been shown to play a role in expected satiety [71] and may present another mechanism by which almonds (and potentially other nuts) may support appetite control. Furthermore, while almonds and the comparator snack were rated as equally palatable, participants rated the almonds as more difficult to chew. The texture and chewiness of almonds may improve their satiating capacity, with evidence suggesting that oral processing plays an important role in food intake by affecting both satiation and satiety [72].

## 4. Overview of Nut Consumption and Body Weight

In order to understand the effects of almonds on appetite control, it is necessary to recognize the complex nature of human appetite as an emergent property of a complex system [73]. The act of food consumption in the real world is influenced by a diverse set of biological and environmental variables, with a greater complexity than can be achieved in laboratory investigations. Recognizing this complexity, it can be shown that changes to the whole diet (for example, by changing energy density) can exert effects on meal sizes, daily energy intake and profiles of hunger [31]. With this in mind, we can ask what is the likely strength of effect on appetite of manipulating a single food in the diet? One systematic review with a meta-analysis of laboratory and field trials has noted that nuts in general do not exert consistent effects on food intake or hunger [52]. However, seeking general effects in an unselected cohort or population will be too insensitive to discriminate effects on people with varying existing degrees of appetite control (satiety phenotypes). An enhancement of satiety is more likely to occur in individuals showing poor appetite control. As shown above, almonds can improve satiety in the low satiety phenotype. This is important since such people are the most likely to benefit from an improvement in control over their appetites (hunger drive and meal size). This action demonstrates how a single food in the diet can exert a meaningful effect. When consumed by people with normal or strong appetite control (high satiety phenotype), the most likely outcome is the maintenance of the habitual eating pattern and a prevention of overconsumption. In achieving these outcomes, almonds (and other nuts) benefit from a range of food factors that influence satiety, including taste and texture, postprandial physiology as well as expectations about satiety. Therefore, in weighing up how nuts can influence appetite control, it is important to manage expectations and not to anticipate the same effect in all types of eaters. Actions can be expected to vary according to the strength of a person’s natural appetite control. Different types of benefit can be expected in people with different forms of satiety control. This approach could form the basis for future investigations of the effect of nuts on appetite.

### 4.1. Evidence from Prospective Cohort Studies

Several prospective cohort studies have examined the association between nut consumption and long-term weight change and obesity risk. Bes-Rastrollo et al. [74] examined the long-term association between nut consumption and weight change over 8 years among 51,188 women aged 20–45 years from the Nurses’ Health Study (NHS) II. The analysis prospectively evaluated the dietary intake of nuts reported in 1989 and subsequent weight changes from 1991 to 1999. After adjusting for age, BMI, alcohol consumption, physical activity, smoking, postmenopausal hormone use, oral contraceptive use and dietary factors, this study found that women who reported eating nuts ≥2 times/week experienced a slightly lower mean (±SE) weight gain (5.04 ± 0.12 kg) than did women who rarely ate nuts (5.55 ± 0.04 kg) (*p*-trend < 0.001). The findings were similar when nut consumption was subdivided into peanuts and tree nuts as well as for participants who are normal-weight, overweight and have obesity.

In an analysis of three prospective cohorts that included 120,877 US women and men with follow-ups ranging from 1986 to 2006 [75], each four-year weight change was inversely associated with a one-serving increment in the intake of nuts (−0.26 kg), fruit (−0.22 kg), vegetables (−0.10 kg), whole grains (−0.17 kg) and yogurt (−0.37 kg), whereas weight gain was positively associated with the intake of potato chips (0.77 kg), potatoes or fries (0.58 kg), sugar-sweetened beverages (0.45 kg), unprocessed red meats (0.43 kg) and processed meats (0.42 kg). These data suggest that specific dietary factors including nuts and overall diet quality influence long-term weight gain.

In a Spanish cohort study consisting of 8865 adult men and women [76], regular nut consumption was significantly associated with a reduced risk of weight gain of ≥5 kg. After adjustment for age, sex, smoking status, physical activity and other covariates, participants who ate nuts ≥2 times/week had a significantly lower risk of weight gain (OR: 0.69; 95% CI: 0.53, 0.90; *p*-trend = 0.006) compared with those who never or almost never ate nuts.

In a prospective analysis of 3092 young adults enrolled in the Coronary Artery Risk Development in Young Adults (CARDIA) study that assessed consumption of walnuts and other nuts three times during the follow-up [77], higher walnut consumption was significantly associated with higher HEI-2015, lower BMI, waist circumference, blood pressure, and triglyceride concentration. Walnut consumers gained less weight since baseline than other nut consumers (*p* ≤ 0.05).

Recently, Li and colleagues evaluated changes in total and different types of nut consumption and long-term weight change in three US cohorts [78]. These analyses included 27,521 men (Health Professionals Follow-up Study, 1986 to 2010), 61,680 women (Nurses’ Health Study, 1986 to 2010) and 55 684 younger women (Nurses’ Health Study II, 1991 to 2011) who were free of chronic disease at baseline in the analyses. The study found that increases in nut consumption, per 0.5 servings/day (14 g), was significantly associated with less weight gain per 4-year interval (*p* < 0.01 for all): −0.19 kg (95% CI −0.21 to −0.17) for total consumption of nuts, −0.37 kg (95% CI −0.45 to −0.30) for walnuts, −0.36 kg (95% CI −0.40 to −0.31) for other tree nuts and −0.15 kg (95% CI −0.19 to −0.11) for peanuts. In addition, increasing the intake of nuts, walnuts and other tree nuts was associated with a lower risk of obesity. In substitution analyses, substituting 0.5 servings/day of nuts for red meat, processed meat, French fries, desserts or potatoes and chips was associated with less weight gain (*p* < 0.05 for all) (Figure 3). This study provides further evidence that increasing daily consumption of total and different types of nuts is associated with less long-term weight gain and a lower risk of obesity in adults. More importantly, this study indicates that replacing “less healthful foods” with nuts may be an effective strategy to help prevent gradual long-term weight gain and obesity.

Nishi et al. conducted a systematic review and meta-analysis of five prospective cohorts on nut consumption and weight gain and obesity among 520,331 participants [79]. It found that higher nut intake was associated with a decrease in overweight/obesity incidence (RR 0.93 [95% CI 0.88 to 0.98], *p* < 0.01; I2 = 90.0%, *p*-heterogeneity < 0.01). Similarly, higher nut consumption was associated with weight loss (MD 0.46 kg [95% CI 0.78 to 0.13 kg], *p* < 0.01; I2 = 95.9%, *p*-heterogeneity < 0.01) and reduced risk of weight gain ≥5 kg (RR 0.95 [95% CI, 0.94 to 0.96], *p* < 0.01; I2 = 46.7%, *p*-heterogeneity = 0.15). The certainty of evidence was rated moderate based on the GRADE criteria. In pooled analyses from models not adjusting for energy intake, higher nut consumption was associated with less weight gain (MD 0.64 kg [95% CI 1.12 to 0.15 kg]).

### 4.2. Evidence from RCTs

Few RCTs have specifically evaluated the role of nuts in weight loss and maintenance or obesity prevention. Wien et al. [80] evaluated the effect of an almond-enriched (84 g/day) or complex carbohydrate-enriched, formula-based, low-calorie diet (LCD) on anthropometric, body composition and metabolic parameters in a randomized 24-week trial among 65 adults with overweight and obesity (age: 27–79 y, BMI: 27–55). LCD supplementation with almonds, compared to complex carbohydrates, led to greater reductions in weight/BMI (−18 vs. −11%, *p* < 0.0001), waist circumference (WC) (−14 vs. −9%, *p* < 0.05), fat mass (−30 vs. −20%, *p* < 0.05), total body water (−8 vs. −1%, *p* < 0.05) and systolic blood pressure (−11 vs. 0%, *p* < 0.02). Ketone levels increased only in the almond-LCD group (*p* < 0.02). This study suggests that an almond-enriched LCD is beneficial for a sustained and greater weight reduction for the duration of the 24-week intervention.

Numerous small, short-term RCTs have examined the effects of nut-rich diets on a wide range cardiometabolic risk factors in which body weight or fatness were secondary outcomes. Fernández-Rodrígue et al. [81] conducted a systematic review and network meta-analysis on the relationship of tree nut and peanut consumption with adiposity measures including body weight (BW), BMI, waist circumference (WC) and body fat percentage (BF%). This study included a total of 105 RCTs with measures of BW (n = 6768 participants), BMI (n = 2918), WC (n = 5045) and BF% (n = 1226). Compared to a control diet, nut-enriched diets had no significant effects on the adiposity-related measures, except for a positive effect of hazelnut-enriched diets and an increase in WC. Moreover, almond-enriched diets significantly reduced WC compared to the control diet. In subgroup analyses with only RCTs designed to assess whether nut consumption affected weight loss, almond-rich diets significantly reduced BMI and walnut-rich diets significantly reduced %BF. This study provides evidence to supports that tree nut and peanut enriched diets do not increase adiposity. A similar conclusion was reached by a meta-analysis conducted by Nishi et al. [79], which found no adverse effect of nuts compared with control diets on body weight (105 trial comparisons involving 9655 participants, MD 0.09 kg, [95% CI 0.09 to 0.27 kg], *p* = 0.34; I2 = 63.2%, *p*-heterogeneity < 0.01).

In a systematic review and meta-analysis of 15 RCTs on almond consumption and cardiovascular risk factors [82], compared to control diets, almond-enriched diets significantly improved blood lipids and reduced inflammatory biomarkers. In the meantime, higher almond consumption of >42.5g/day significantly improved fasting blood glucose and reduced BMI.

In a systematic review and meta-analysis of 55 parallel-arm or crossover interventions of nuts (including mixed nuts, nut-based snack bar and individual nuts including almonds, cashews, hazelnuts, macadamia nut, peanut, pecan, pistachio and walnut) there was no change reported in body weight, BMI or waist circumference. The mean duration of these studies was 13.8 ± 21.5 weeks and the mean intake of nuts was 48.2 ± 20.8 g/d. The analysis included studies where no substitutions instructions were provided as well as studies which provided to the participants instruction on substitution. In the studies where substitution instructions were not provided, there was no change in body fat percentage. In studies with dietary substitution instructions, there was a significant decrease in body fat percentage [83].

Only one long-term RCT examined the effects of a Mediterranean diet supplemented with nuts on body weight and waist circumference changes in the context of a Mediterranean dietary intervention and primary prevention of CVD [84]. The PREDIMED trial randomly assigned 7447 participants with high risk of CVD to one of three interventions: Mediterranean diet supplemented with extra-virgin olive oil (n = 2543); Mediterranean diet supplemented with mixed nuts including almonds, walnuts and hazelnuts (n = 2454); or a control diet (advice to reduce dietary fat; n = 2450). After a median 4.8 years of follow-up, participants in all three groups had marginally reduced bodyweight. After multivariable adjustment, the difference in 5-year changes in bodyweight in the olive oil group was −0.41 kg (95% CI −0.83 to 0.01; *p* = 0·06) and −0.02 kg (−0.45 to 0.42; *p* = 0.94) in the nut group compared with the control group. The adjusted difference in 5-year changes in waist circumference was −0.47 cm (−1.11 to 0.18; *p* = 0.15) in the olive oil group and −0.92 cm (−1.60 to −0.24; *p* = 0.008) in the nut group compared with the control group. This study provides strong evidence that diets supplemented with either extra-virgin olive oil or nuts had no adverse effects on body weight or WC. In contrast, these diets may have beneficial effects on adiposity measures compared to a lower-fat diet.

### 4.3. Methodological Issues in Observational Studies and RCTs

Both observational studies and RCTs of diet and body weight fraught with methodologic problems (see Chapter 14 in [85]). RCTs should provide some of the most rigorous evaluations of dietary intake and body weight. However, long-term dietary intervention studies are seldom feasible because of the high cost and lack of compliance by study participants. In addition, lack of compliance and high dropout rates are common in dietary intervention trials. Most RCTs on body weight and other CVD risk factors are of short-duration, small sample sizes and use different control groups. In most RCTs, adiposity measures such as weight or waist circumference changes were considered as secondary outcomes.

Observational studies of nut consumption and body weight are also complicated by several methodologic issues. In particular, residual confounding by other dietary and lifestyle factors cannot be ruled out because regular nut consumers tend to follow a healthier diet and lifestyle than non-consumers. Dietary assessment tools such as the 24 h recalls, dietary records and FFQs that are widely used in epidemiologic studies are prone to both random and systematic measurement errors. Although carefully validated FFQs that are administered repeatedly during follow-up are best-suited to the assessment of long-term patterns in intake, few large-cohort studies assessed diets repeatedly. In addition, no study has specifically examined the influence of food processing methods on body weight outcomes (i.e., salted, raw, roasted). Finally, most studies have been conducted in white or European populations, and thus the results may not be generalizable to other racial and ethnic groups.

## 5. Potential Components of Nuts That Contribute to Weight Control

Several mechanisms have been proposed to explain the potential benefits of nut consumption on body weight [86] (Figure 4). Nuts are rich in (1) proteins and (2) dietary fiber, which are associated with increased satiety, and in (3) unsaturated fats, which may increase oxidation that potentially decreases body fat accumulation [87]. High amounts of protein and fiber in nuts may also increase thermogenesis and resting energy expenditure. Dietary fiber (especially viscous fiber) in nuts delays gastric emptying and subsequent absorption that potentially suppresses hunger and promote healthy gut microbiome that improves energy metabolism. In addition, incomplete mastication of nuts may lead to increased energy loss via feces, which contributes to energy availability of nuts and thus a lower energy intake. Furthermore, consuming nuts at expense of red meat and refined carbohydrates may also contribute to less weight gain and lower risk of chronic diseases. 

## 6. Clinical and Public Health Dietary Recommendations on Nuts and Weight Management

Cumulative evidence from long-term large cohort studies supports that an increased consumption of nuts, including total nuts and different types of nuts, is associated with less weight gain and lower risk of obesity, despite being calorically dense. The benefits to body weight are more pronounced when nuts are used to replace unhealthy foods such as red meat, processed meat, French fries, desserts or potato, chips. In addition, short-term RCTs suggest that nut-enriched diets had no adverse effects on body weight or other adiposity measures compared to control diets. There is some evidence that nuts may have beneficial effects on weight loss and maintenance, although more research is needed. Healthy dietary patterns rich in nuts, such as the Mediterranean diet, DASH diet and healthy plant-based diets, have been associated with age-related weight gain, although in these studies, the effects of nuts cannot be separated from other components of the dietary patterns [84].

## 7. Conclusions

To date, the plant cell wall factors that influence the energy available from nuts have mostly been investigated in almonds, with some research conducted in pistachios and walnuts. The effect of the plant cell wall, and its fermentation, on energy availability of other nuts has not been reported. Furthermore, the metabolizable energy value of nuts has been measured for almonds, walnuts, pistachios and cashews. Data from other nuts have not been reported. Additionally, the effect of processing on energy availability has only been investigated in almonds and peanuts. More information on dose response and individual variability may be useful to understand individual variability in energy intake, especially when trying to determine compensation of energy intake.

Evidence from RCTs and observational cohorts indicates higher nut consumption does not appear to cause greater weight gain; rather, nuts may be beneficial for weight control and prevention of long-term weight gain. Diet and lifestyle changes such as the replacement of less healthful food items (e.g., red or processed meats, refined grain products) with nuts and other healthy foods have the potential to reduce risk of obesity and obesity-related chronic diseases. In terms of future directions, more observational studies and RCTs are needed to examine the effects of nut consumption on different body depots, especially abdominal, visceral and liver fat. More studies are also needed to be conducted among individuals with type 2 diabetes, the metabolic syndrome and fatty liver disease and in diverse populations of different racial and ethnic groups and socio-economic status. Finally, research is needed to examine the role of nuts in healthy and sustainable eating patterns such as the Healthy Planetary Diet recommended by the Eat-Lancet Commission [88].

## Figures and Tables

**Figure 1 nutrients-15-01162-f001:**
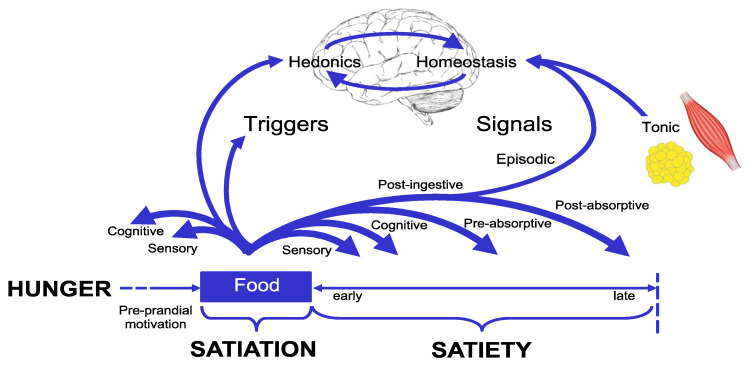
The Satiety Cascade depicts meal size and the time between meals is influenced by the processes of satiation and satiety. It also demonstrates the interaction between the homeostatic and hedonic influences on the processes of satiation and satiety. Adapted from Blundell and Finlayson [30].

**Figure 2 nutrients-15-01162-f002:**
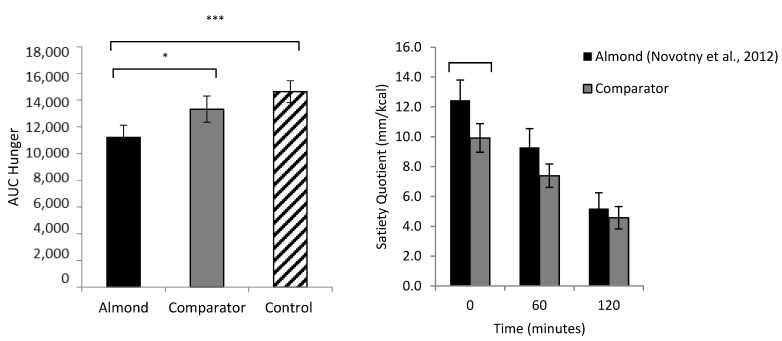
Left: Area under the curve hunger for the almonds condition, energy- and weight-matched comparator (crackers) and weight-matched comparator (water). Right: Satiating efficiency (measured by the Satiety Quotient) of the almonds compared to comparator for 120 min post-consumption [16]. Adapted from Hollingworth et al. [61]. Note: * *p* < 0.05; *** *p* < 0.001.

**Figure 3 nutrients-15-01162-f003:**
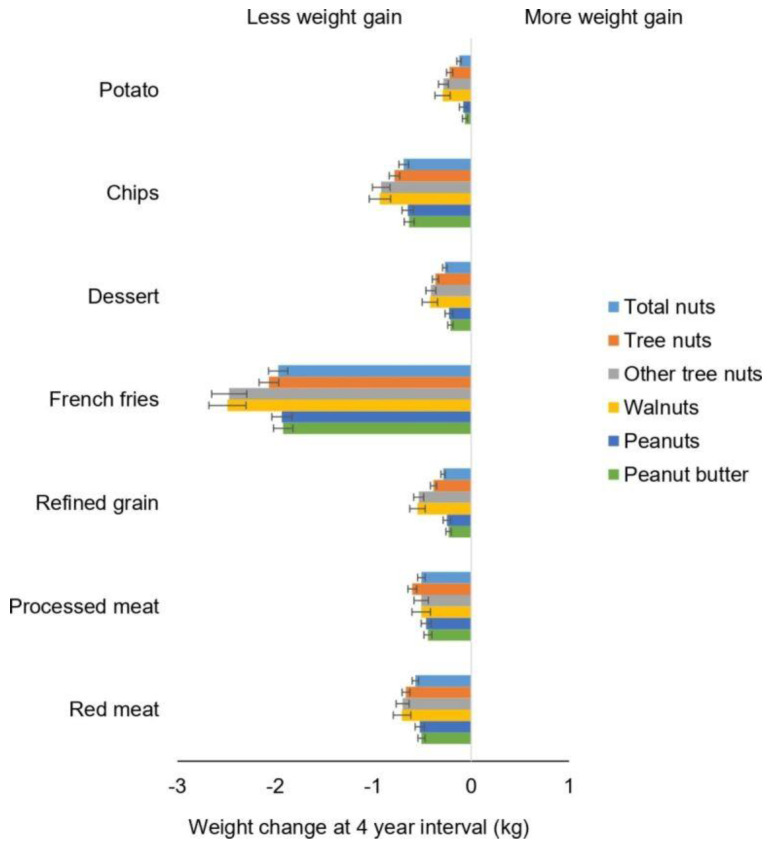
Association between weight change (kg) every 4 years and substitution of nuts and individual types of nuts, per 0.5 servings/day with equal serving of other food items among NHS, NHS II and HPFS. Weight changes are presented as solid bars; T bars represent 95% CI. Multivariate model was adjusted for age, menopausal status (pre- or postmenopausal) and hormone therapy use (never, past or current) in women; baseline BMI of every 4 years; hours of sleeping at baseline; changes in lifestyle factors: smoking status (never, former, current: 1 to 14, 15 to 24, or ≥25 cigarettes/day), physical activity (MET hours/week), hours of sitting (hours/week); and changes in dietary factors: fruits, vegetables, alcohol, snacks, dessert, French fries, red or processed meat, whole grain, refined grain products and sugar sweetened beverages.

**Figure 4 nutrients-15-01162-f004:**
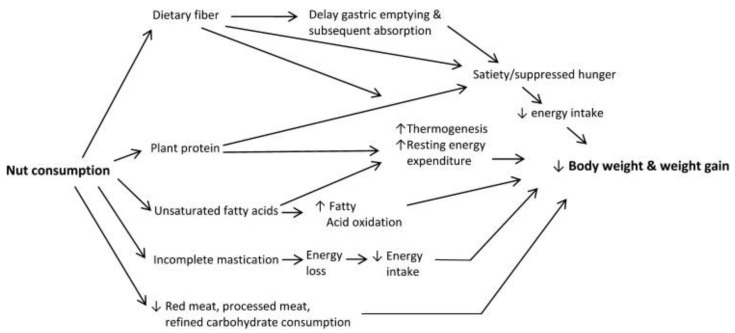
Conceptual framework of potential mechanisms linking nut consumption to decreased body weight and weight gain [86].

## Data Availability

Not applicable.

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
