# Peer review of "Nuts, Energy Balance and Body Weight"

_nutrients, 2023, doi:10.3390/nu15051162_

Round 1
Reviewer 1 Report
This is a comprehensive review of nuts and issues relating to energy balance and weight/adiposity. I have a few minor suggestions:
Could you briefly summarize the key conclusions of the review in the abstract?
Please use person-first language throughout the paper (i.e., adult with obesity) and describe individuals or populations as having overweight or obesity not being overweight or obese.
Line 96: Revise to “The number of particles recovered…”
Line 201-03: Add referencing for these sentences.
Line 419: For fruit, convert pounds to kg to be consistent.
This meta-analysis could be cited in section 4.2: Advances in Nutrition, Volume 12, Issue 2, March 2021, Pages 384–401, https://doi.org/10.1093/advances/nmaa113
Author Response
Could you briefly summarize the key conclusions of the review in the abstract? Added:
Consistently, the evidence from RCTs and observational cohorts indicates higher nut consumption does not appear to cause greater weight gain; rather, nuts may be beneficial for weight control and prevention of long-term weight gain. Multiple mechanisms likely contribute to these findings including aspects of nut composition which affect nutrient and energy availability as well as satiety signaling.
Please use person-first language throughout the paper (i.e., adult with obesity) and describe individuals or populations as having overweight or obesity not being overweight or obese. Obese was changed to obesity.
Line 96: Revise to “The number of particles recovered…” - Done
Line 201-03: Add referencing for these sentences. References added:
- Cecil, J.; Dalton, M.; Finlayson, G.; Blundell, J.; Hetherington, M.; Palmer, C. Obesity and eating behaviour in children and adolescents: contribution of common gene polymorphisms. Int Rev Psychiatry 2012, 24, 200-210, doi:10.3109/09540261.2012.685056.
- Blundell, J.E.; Stubbs, R.J.; Golding, C.; Croden, F.; Alam, R.; Whybrow, S.; Le Noury, J.; Lawton, C.L. Resistance and susceptibility to weight gain: individual variability in response to a high-fat diet. Physiol Behav 2005, 86, 614-622, doi:10.1016/j.physbeh.2005.08.052.
- Aaseth, J.; Ellefsen, S.; Alehagen, U.; Sundfor, T.M.; Alexander, J. Diets and drugs for weight loss and health in obesity - An update. Biomed Pharmacother 2021, 140, 111789, doi:10.1016/j.biopha.2021.111789.
- Jeong, D.; Priefer, R. Anti-obesity weight loss medications: Short-term and long-term use. Life Sci 2022, 306, 120825, doi:10.1016/j.lfs.2022.120825.
- Blundell, J.E.; Finlayson, G. Is susceptibility to weight gain characterized by homeostatic or hedonic risk factors for overconsumption? Physiol Behav 2004, 82, 21-25, doi:10.1016/j.physbeh.2004.04.021.
Line 419: For fruit, convert pounds to kg to be consistent. Units corrected (the value of -0.22 is correct – units needed to be changed).
This meta-analysis could be cited in section 4.2: Advances in Nutrition, Volume 12, Issue 2, March 2021, Pages 384–401, https://doi.org/10.1093/advances/nmaa113. Reference has been added.
Reviewer 2 Report
Review of the article „Nuts, Energy Balance and Body Weight“
The studied topic is very interesting and I support research on this topic because many studies have shown that nuts can reduce the risk of chronic diseases. This review paper presents the various factors associated with energy intake from nuts, including the food matrix and its influence on digestibility, and the role of nuts in appetite regulation. The article tried to show that the high energy value of nuts does not affect the increase in body weight, as some previous studies have shown.
As stated in the paper, the metabolic energy value of nuts was measured for almonds, walnuts, pistachios, and cashews, while it was not for other nuts. Also, the processing effect was not tested for all nuts. Regardless of some shortcomings of this work, I think it is very good work.
I haven't specific corrections and suggestions.
Author Response
Thank you!